# Dealing with Teacher Shortage in Germany—A Closer View of Four Federal States

Sandra Seeliger [1] and Marcia Håkansson Lindqvist [2,*]

1. School of Human and Social Sciences, University of Wuppertal, 42119 Wuppertal, Germany
2. Department of Education, Mid Sweden University, 85170 Sundsvall, Sweden
* Correspondence: marcia.hakanssonlindqvist@miun.se

**Abstract:** Teacher shortage can be said to be an international challenge. In this article, the phenomenon of teacher shortage in 4 of the 16 German states is explored and analysed. These four states were selected to demonstrate the variety and give examples of different approaches dealing with teacher shortage. Similarities and differences were observed and the most evident similarity is that all states that are presented have to deal with a shortage of teachers. Fluctuations of student numbers and waves of retirements, and increased need of new employment may be foreseen. However, the resources of the universities and teacher mobility between states is not always facilitated. The number of side entrants is remarkably low in some states. This may mean that their planning strategies are better or their recruitment for teacher training study places is more successful than in those states that need to employ side entrants on a regular basis. These are questions that cannot be answered yet. In conclusion, more research is called for to study how these results can be interpreted in order to allow the findings and their implications to be discussed in the broadest context possible.

**Keywords:** teacher shortage; education; student numbers

## 1. Introduction

Education is a fundamental human right and the right of every child on which our will be built. [1] It "works to raise men and women out of poverty, level inequalities and ensure sustainable development. Worldwide 258 some million children and youth are still out of school for social, economic and cultural reasons. Education is one of the most powerful tools in lifting excluded children and adults out of poverty and is a stepping stone to other fundamental human right and education can be seen as the most sustainable investment. The right to quality education is already firmly rooted in the Universal Declaration of Human Rights and international legal instruments, the majority of which are the result of the work of UNESCO and the United Nations. [2]" To ensure every child education, many important resources are needed, such as school buildings, a safe way to school, and material security. One of the most important factors for student learning is the teacher [3,4]. This means that students need access to competent and motivated teachers. However, teacher shortage appears to be a challenge worldwide. The United Nations (UN) report the need for some 69 million new teachers in order to achieve the targets set in Agenda 2030 [5]. In regard to teacher shortage, attracting and retaining new teachers is difficult in certain geographical areas, as well as in certain subject areas.

Teacher shortage is a common challenge for many developed countries [6], and it is seen in Europe [7,8]. In England, shortages are expected to increase as the result of a larger number of students, a phenomenon simultaneous to an increase regarding teachers who retire [9,10]. Teacher shortage is also seen in the Nordic countries, for example in Sweden [11]. Teacher shortage also appears to be a challenge in the US [12,13].

The aim of this article was to explore and analyse the situation in four German states regarding the phenomenon of teacher shortage. The research questions were: (1) What

is the current status of teacher shortage in schools in these German states? (2) What are the main aspects impacting teacher recruitment and retention in Germany? (3) What similarities and differences can be seen?

Through document analysis, the study will deepen the understanding of the current status in relation to teacher shortage and the impacting factors in Germany. Following a short introduction of the overall situation in Germany, specific cases will be presented.

## 2. Teacher Shortage in Germany

Germany is a federal state. The country is a union of sixteen states, each is responsible for its education and schooling. Every state tries to find the best solutions for teaching, schooling and day care. Teacher shortage is a phenomenon most states must deal with. However, there are differences. The shortage of teachers does not equally affect all subjects, all school types, or all regions. Teacher associations, parents, media, and politicians repeatedly point to the lack of teachers.

Currently, the existing and expected shortage of teaching staff at primary schools is a cause for concern throughout Germany. This concern increases regarding the recent political situation; in February 2022, Russia launched a war of aggression against Ukraine. Many people, especially women and children (many of them at the age of 6 to 10 years, which means they will join primary schools), fled to neighbouring European countries. Between 24 February and 6 April 2022, around 310,000 war refugees from Ukraine were recorded to enter Germany [14]; by October, 2022 the number increased to more than one million people (1.008.935, cp. ibid). The children among the refugees are going to join schools that are probably already at their limit with teachers who are also at their limit. Nevertheless, especially these children need welcoming schools and ambitious teachers.

In the school year 2019/2020, there were 32,332 schools for general education in Germany, with a total of 782,613 teachers and 8,326,884 pupils [15–17]. Two years later, in the school year 2021/2022 there were 32,206 schools for general education in Germany with a total of 708,962 teachers and 8 432,386 pupils [18]. The numbers show that the situation became worse in the last two years and is expected to become even worse, taking into consideration that the effects of the Ukraine war are not fully visible in the numbers of the school year 2021/2022.

Each of Germany's 16 federal states has its own federal government and its own school law. The individual states are historically responsible for education and schooling, but of course all states are bounded to constitutional law. In order to have some sort of consistency and more transparency, the Standing Conference of Ministers of Education and Cultural Affairs (KMK) was founded in 1948 [19]. The KMK plays a significant role as an instrument for the coordination and development of education in the country.

In October 2018, the KMK calculated a shortage of 15 300 primary school teachers in 2025 [20], this was a dramatic finding, which was confirmed and seen as even more dramatic by researches of the Bertelsmann Foundation. Their study refers to the Federal Statistical Office's population forecast from June 2019. For the year 2025, it predicts that there will be 3.254 million to 3.323 million children between the ages of 6 and 10 in Germany, which leads to an estimated lack of 26 304 primary school teachers [21].

Since then, the KMK has been publishing the expected teachers' shortage on an annual basis. For this article, the numbers from March 2022, which give a forecast till 2035, are compared with the actual teaching situation in four German states.

In summary, the federal structure in Germany leads to a variety of different school systems, teacher training programs and alternative programs to become a teacher. It can be stated that the teacher shortage cannot be denied and can be considered to have negative effects on both teachers and pupils.

## 3. Theoretical Framework

The theoretical framework used in this article is inspired by the policy enactment theory [22]. This assumes that stakeholders, such as researchers, policyholders, producers

of national statistics, and the media, interpret, reinterpret, and use policies. Stakeholders' actions and perceptions are, therefore, formed by their specific contexts. These actions and perceptions are continually reconstructed in context and through context as policy is realised in practice. This process can be described as a dialectic process, which involves interpretation and reinterpretation. Such is the case in educational policy, which is interpreted and re-interpreted as activities and processes, both of which, intentional and unintentional and planned and unplanned, are enacted. Here, stakeholders become enactors of policy in policy development. Policy enactment theory, thus, provides the possibility to see how stakeholders' relations between policy and practice may be seen in light of how educational policy is interpreted, represented, and re-contextualised. In other words, policy enactment theory offers the possibility to capture stakeholders' perceptions of teacher shortage and provide deeper insight in highlighting teacher shortage as a phenomenon. With a base of policy enactment, the data materials were then grouped into themes inductively in a reflexive thematic analysis [23]. Thus, this framing aspires to contribute to the exploration and analysis of teacher shortage in Germany in four specific states and explores some of the similarities and differences between these four states.

## 4. Materials and Methods

Inspired by policy enactment theory [22], a description of teacher shortage in Germany was constructed. The research design involved the analysis of documents on teacher shortage. The relevant documents were identified through academic databases, national statistics, media and the internet (FIS-Bildung, Destatis, Google, and Google Scholar). For the German states statistical publications of the KMK, deliver highly relevant data about 'expected teachers to be employed' in Germany from 2018 to 2030 (2035). The calculation is based on demographic figures and trends. Thus, the empirical base for this study was publications in media, scientific articles, policy documents, reports, books, and statistics. Thus, the data materials were collected using open accessible articles on the internet, such as local press, homepages of the states and statistical offices, which were thereafter analysed.

A comparative study of teacher shortage in Germany is needed for several reasons. First, it can identify the mediating role of cultural differences regarding teacher shortage. Second, it could provide fruitful new contributions in understanding teacher shortage. Thus, the findings of this study may contribute to the relevant knowledge in other countries facing challenges with teacher shortage.

As an intensive, holistic description of cases, this study attempts to maintain the distinctive attributes of a case study. A case study should be descriptive, heuristic, and interpretivist [24–26]. The study focuses on the particular situation concerning teacher shortage in the four specific federal states in Germany. This article attempts to provide a rich description of teacher shortage in Germany based on different stakeholders' perceptions, so the study can be said to be descriptive. Further, the study aspires to explore and highlight the understanding of teacher shortage based on the contexts of the different states. In this study of teacher shortage in the different German contexts, the cases demonstrate similarities and differences, for example, regarding infrastructure, statistics of schools, and teachers, as well as the perceptions of the stakeholders.

## 5. Results

This section will provide a closer look of the selection of four German states and how they are affected by and deal with teacher shortage. The results were presented for four German states: Bayern, Berlin, Brandenburg, and Nordrhein-Westfalen. A closer look on these four German states will be taken to highlight the specific problems caused due to the federal structure, regional characteristics, the variety in size, population, population density, prosperity and teaching situation. The selection was based on the idea of selective sampling [27], which led to an exemplary and contrasting description of the issues, measures, and strategies in dealing with teacher shortage.

### 5.1. Bayern

Bayern is located in the south of Germany. In terms of territory, it is the largest state in the Federal Republic of Germany, with a size of 70,541 square kilometres. Around 13.076 million people live in Bayern, which leads to a population density of 186 people per square kilometre in 2021 [28] see Table 1.

From 2014 to 2018 there was no teacher shortage in Bayern. In 2019, the territory was short of 340 teachers; in 2020 the shortage increased to 900. Primary schools and lower secondary schools were most affected (cp. [29,30]). In 2021, Bayern/Bavaria was able to hire 5000 people for regular teaching positions. Due to the pandemic situation, some teachers, such as pregnant teachers, were not allowed to teach face to face, which led to an additional 800 positions for so-called team teachers who were in local schools and there supported by the home-staying teachers [31].

As teacher shortage finally started to affect Bayern, political action attempted to implement several measures for more people to obtain teaching qualifications, most important among these is an additional qualification for secondary school teachers to be able to teach at primary schools. Since the school year 2021/2022, middle-school teacher training is available for graduates of German, German as a foreign language, English, computer science, biology, physics, chemistry or mathematics as major subjects [31]. From September 2022 onwards, this way of teacher training will also be available for graduate (Master of Science, Master of Arts, Diploma) students of physics, computer science, and arts with the aim to teach at high schools (Gymnasium) [32]. Other suggested measures, such as refusing part-time positions or not offering sabbaticals, are restrictive and result in resistance. Parents are used to properly qualified teachers for their children and teachers are not willing to follow the voluntary suggestions of the Ministry for Education. They point out that their working conditions are already hard, and they work overtime, so enlarging classes or delaying the age of retirement will not fix the problem [33,34].

Furthermore, the Ministry of Education responded by launching a modern and interactive advertising campaign for the teaching profession. Information about the teaching degree program is clearly presented, as well as the advantages of the profession (salary, opportunities, and security) are highlighted [35].

In short, Bavaria has only recently had to deal with a shortage of teachers. The options to open teacher-training programs for university graduates with subjects in related degrees (Master of Science, Master of Arts, Diploma) are only used when positions cannot be filled. This means that delays in the process of filling positions can be expected.

### 5.2. Berlin

Berlin is a popular and growing city. Berlin is the capital of Germany, a state of its own, and sees itself as a loveable and liveable place. With around 3.7 million inhabitants, Berlin is the most populous city in Germany. A total of 19.6 percent of Berlin's inhabitants are foreigners, which makes Berlin the state with the highest percentage of people with a non-German nationality.

Berlin has a size of 891 square kilometres; the population density is very high (4090 inhabitants per square kilometre, see Table 1). Despite the population density, 40 percent of Berlin is marked as woodland, agricultural areas, water bodies, gardens, or parks; another 13 percent of Berlin's outdoor area is reserved for recreation, sports, and leisure time activities. A dense childcare network, 1274 schools, more than 1800 playgrounds, and a range of support services for parents and children, make Berlin attractive for families [36].

Therefore, Berlin needs many teachers. However, Berlin has difficulties getting enough teachers for vacant positions. In the past, Berlin decided to be more flexible as an employer for teachers and stopped employing teachers as civil servants in 2004. Therefore, schools could pay more flexible wages or dismiss unprofessional teachers, but mostly it was a strategy to save money, as becoming a civil servant is attractive for many teachers in training, graduate teachers prefer to apply for a position at a school in a state which does employ them as civil servants. So, instead of improving the teaching quality at Berlin's

schools, this policy led to a lack of applicants, and therefore, to worse quality in schools. As a result of these developments, the Teachers' Associations and politicians aimed for a return to tenure. Since the school year 2022/23, all newly trained teachers are being offered employment as civil servants in the state of Berlin in accordance with the applicable civil service regulations [37]. This regulation allows Berlin to invite teachers who are civil servants in other states to apply for a position in Berlin, which enables national teacher mobility.

Dealing with a severe shortage, Berlin hires academics who have studied a subject that is related to school subjects and qualifies them, while they are already teaching. Recently, there are districts in Berlin where 60 to 70 percent of new teachers hired are 'Quer-einsteiger' (side entrants) [37].

According to the Berlin Senate Education Department, there were 2440 permanent teaching positions to be filled for the school year 2021/22, which was nearly 500 less than KMK expected in 2020. At the beginning of the school year, 2886 teachers could be hired, but not all of them full-time. A total of 1526 of them were applicants of the teacher training program. According to the Senate Education Department, 150 applicants were still in the recruitment process at the start of the school year, and 80 positions were still vacant. Regardless of the vacancies, there were already 1360 new hires of career changers in the school service at the beginning of the school year. The Union for Education and Science (Gewerkschaft Erziehung und Wissenschaft) assumes that a total of 60 per cent of newly recruited teachers this school year will be recruited as side entrants [31].

Some of Berlin's politicians were pushing to re-establish teachers' status as civil servants [38] and recent elections made this option work. However, it is also unclear whether it is the teachers' status or the demanding work in Berlin's "hot spots" that repel junior teachers from Berlin's schools.

### 5.3. Brandenburg

Berlin and Brandenburg are geographically melted together. Berlin as the capital of Germany is a state of its own and is located in the middle of the state Brandenburg. The example of Berlin and Brandenburg demonstrates how the teaching situation in some states is linked together and what problems different political strategies can cause.

The state of Brandenburg is located in the north east of the Federal Republic of Germany. With a total area of 29,476 square kilometres, Brandenburg is one of the most extensive states in Germany. The landscape of Brandenburg is characterised by lowlands and valleys, with numerous rivers and lakes. A total of 33,000 km of river area, including the Spree, Havel, Rhin, Nuthe and Dahme, as well as a multitude of canals and around 3000 lakes, makes Brandenburg the state with the most water in the Federal Republic of Germany [39]. With 2,511,917 inhabitants and a population density of 85, Brandenburg is much less populous than Berlin (see Table 1).

In the summer of 2021, Brandenburg was able to fill all vacant teaching positions, most of them with fully qualified teachers; 256 of the 1256 positions were filled by side entrants, that is a percentage of 21.3. In 2020, the percentage of side entrants was 32.5. All in all, in 2021, 2700 side entrants taught at Brandenburg's schools, that sums up to a percentage of 13.1 of all teachers. In Berlin, about 60 percent of the newly employed teachers are side entrants [31].

Compared to Berlin, Brandenburg has a quite comfortable teaching situation, even though the KMK calculated that the expected shortage of teachers would sum up to 655. At primary schools, 560 teachers were needed in 2021, but only 120 were expected to finish their teacher training (see Table 2, based on [30,31]). As universities offer a double qualification for primary and some secondary schools, the surplus of those graduates may reduce the shortage. This, however, requires attractive payment and working conditions. Brandenburg offers equal payment to primary school teachers, secondary school teachers, and special education teachers. Working full time means, in Brandenburg, to teach 27 h at a primary school, or 25 h at any secondary school, which is less than in most other states. At

least when they start teaching at schools, all teachers are paid equally [40]. So, Brandenburg and its responsible ministers feel well-prepared for teacher shortages.

Additionally, as the recent report about the teaching situation at Brandenburg's schools shows that all positions could be filled, they might be right. However, there are critical voices too. The teachers' association of Brandenburg [41] points out that there is only one university in Brandenburg offering teacher training and that teacher shortage in other states of Germany may affect the schools in Brandenburg as well. So, maybe in future, Brandenburg will not be able to recruit teachers from Berlin or other states.

### 5.4. Nordrhein-Westfalen (NRW)

"Around 17.9 million people live in NRW (as of 31 December 2020). This makes NRW the most populous of Germany's 16 states. At the same time, NRW is also the most densely populated state in Germany. Around 525.49 inhabitants live here per square kilometer. With an area of 34,112,4 square kilometers, NRW is the fourth largest state in Germany. It borders the states of Niedersachsen, Hessen and Rheinland-Pfalz as well as Belgium and the Netherlands. Furthermore in 2020 NRW generated around 697.1 billion euros (almost 20.9 percent) of Germany's gross domestic product (GDP)" [42].

In NRW, there are 5128 schools in total; among these, 2784 are primary schools (see Table 1, [15,43]). In June 2021, 3662 teaching positions for the coming school year still were vacant, and one year later, in June 2022, 4369 positions were vacant [31,32].

In 2018, NRW faced a shortage of 7640 teachers; in 2020 the calculated shortage was 370. It has to be pointed out that the shortage of 2018 was not covered then [20]. So, by 2022 the shortage was added up to almost 4400 vacant teacher positions, which means that a percentage of 2.7% of the teaching positions were not filled in NRW [44]. In a factsheet, the ministry of school and education summarises that the situation in primary schools, schools for students with special needs, and lower secondary schools, as well as at schools for vocational education, is tense. In sum, there will be a shortage of 15,000 teachers over the next ten years at those schools but all in all a surplus of 16,000 teachers for grammar schools and comprehensive schools is expected [45]. Thus, many school leavers want to become teachers, but they reach for the best paid teaching positions that NRW's school system is offering. The Ministry of School and Education suggests a package of different measures to motivate teachers to apply for teaching positions at schools that are struggling to fill their vacancies. Firstly, there is a possible additional charge of up to EUR 350. Secondly, side entrances are made more attractive; becoming a teacher based on regular university education should be made more attractive by taking vocational experiences into account to quantify the salary. Thirdly, already retired teachers are asked to improve their annuity by teaching lessons at school again [45].

Another measure focuses on teacher training. The government is forcing an expansion of study capacities for primary school teachers and teachers of students with special needs. Recently, 700 additional students can subscribe for a Bachelor of Education for primary schools, and 750 additional students can apply for a Bachelor of Education in special needs [45].

As inclusive education is expanding in Germany, more schools need specialised teachers than before. To fill these vacancies, regular teachers can apply for these jobs and qualify "on the job"; former more restrictive rules were loosened up [45].

Further measures aim for the efficient use of resources, such as re-employing retired teachers and motivating teachers in parental leave to resume teaching as soon as possible by offering a teaching position at a school close to their home when they only stay home for 8 instead of 12 months [45].

Furthermore, the government is trying to improve the image of the teaching profession by media campaigns. In 2019 and 2020, one million euros each year were planned to be spent on campaigns to obtain more teachers [45].

In addition to the problems in filling vacant positions at schools, in 2022, the minister of school and education of NRW, Yvonne Gebauer, announced that 4000 additional

positions for schools in NRW would be established. These positions do not only include teaching positions but also positions for other pedagogical experts (i.e., for teaching in multi-professional teams and improvements in inclusive teaching) [46]. Implementing additional pedagogic positions may also affect willingness to become a teacher.

In NRW, the range of measures to improve the teaching situation at schools is wide; this shows how eager politicians are to find a solution. The competition among German states regarding education is probably one of the pushing factors.

### 5.5. Results in Summary

In summary, the results show differences and similarities between the four states. Bayern, for example, has a lower population density and larger rural area. In this area, the rules for teacher training are very different to other states. This makes it difficult for graduate teachers from other states to teach in Bayern, as well as for teachers from Bayern to teach in other German states. In contrast to Nordrhein-Westfalen, Bayern has little experience with teacher shortages and seems to have tried and tested only a few strategies available so far. However, Nordrhein-Westfalen, with almost 18 million inhabitants, is one of the most populous states and has the highest population density. This state has a high level of teacher shortage. Here, there is a distinction between immediate, short-term measures and long-term solutions, as well as a large variety of specific measures. Berlin currently needs 1149 teachers. As already noted, Berlin's teachers were not employed as civil servants (which recently changed from the school year 2022/23 onwards), which was, and still, is one of the most important benefits for most teachers in training. Returning to the status of civil servant may end the ambitions of many fully trained teachers to get a position in Brandenburg. Nevertheless, Brandenburg copes with teacher shortage by employing side entrants. Around 30.1 % of all newly hired teachers in Brandenburg in the school year 2022/23 did not undergo traditional teacher training; in the years before, the percentage was 20.9 (2021) and 32.5 (2020). Whether this approach holds more opportunities than risks remains to be evaluated. This summary is illustrated in Table 1.

As illustrated in Table 2, there will be a teacher shortage in the near future, and even ten years in future, there will be a teacher shortage in several school types. In 2026, there will be an overall shortage of teachers for all school types in every state presented in this article. In 2034, the situation can be expected to improve. However, until then, schools and politics have to ensure adequate schooling and teaching conditions for students and teachers regardless.

**Table 1.** The presented states in numbers (own presentation based on [47,48]).

| State | Size in km² | Inhabitants 2018-12-31 | Population Density 2018-12-31 (Inhabitants per km²) | Expected Shortage in 2022 (Own Calculation Based [49].) | Teaching Situation in 2022/2023 (Mainly Based on [32].) | Number of Schools in Total (September 2022) | GDP 2020 in EUR m ([48]) |
|---|---|---|---|---|---|---|---|
| Bayern | 70,541.57 | 13,076,721 | 185 | 1850 | 4000 newly employed teachers, side entrants among them, about 500 vacant fulltime positions | 4647 | 610,217 |
| Berlin | 891.12 | 3,644,826 | 4090 | 1.149 | 1610 newly employed teachers, 455 side entrants among them, still about 875 vacant full-time positions | 1288 | 154,634 |
| Brandenburg | 29,654.48 | 2,511,917 | 85 | 560 | 1322 newly employed teachers, 387 side entrants among them, still 63 vacant positions | 1448 | 73,931 |
| Nordrhein-Westfalen | 34,112.31 | 17,932,651 | 526 | Surplus of 720 (in 2022, there was a shortage of 863 teachers in primary schools and a shortage of 498 teachers in lower secondary schools. On the other hand, there is a surplus of 1936 teachers for upper secondary schools.) | 4369 vacant positions in August 2022, severe shortage at primary schools | 5102 | 697,125 |

**Table 2.** Difference between supply of teachers and demand for teachers—shortage at minus (own calculation based on [49]).

| School Type | Year | Bayern | Berlin | Brandenburg | Nordrhein Westfalen |
|---|---|---|---|---|---|
| primary schools | 2022 | −450 | 15 | −155 | −863 |
| | 2026 | 670 | 14 | −110 | 394 |
| | 2030 | 800 | 14 | −60 | 727 |
| | 2034 | 1140 | 14 | −60 | 554 |
| lower secondary schools | 2022 | −670 | −589 | −260 | −498 |
| | 2026 | −670 | −525 | −165 | −761 |
| | 2030 | −730 | −507 | −95 | −733 |
| | 2034 | −310 | −507 | −95 | −389 |
| upper secondary schools and gymnasium | 2022 | −350 | −89 | 10 | 1936 |
| | 2026 | −420 | −80 | −100 | −2876 |
| | 2030 | −480 | −78 | 45 | 1235 |
| | 2034 | −520 | −78 | 45 | 1245 |
| vocational schools | 2022 | −180 | −236 | −65 | 41 |
| | 2026 | −140 | −211 | −100 | −282 |
| | 2030 | −210 | −204 | −35 | −628 |
| | 2034 | −230 | −204 | −35 | −748 |
| teachers for students with special needs | 2022 | −300 | −223 | −100 | 104 |
| | 2026 | −190 | −198 | −85 | −85 |
| | 2030 | −40 | −191 | −70 | 353 |
| | 2034 | 120 | −191 | −70 | 415 |
| total—all school types | 2022 | −1950 | −1122 | −570 | 720 |
| | 2026 | −750 | −1000 | −560 | −3610 |
| | 2030 | −660 | −966 | −215 | 954 |
| | 2034 | 200 | −966 | −215 | 1077 |

## 6. Discussion

The aim of this article was to explore and analyse the situation in four German states regarding the phenomenon of teacher shortage. The research questions are: (1) What is the current status in relation to teacher shortage in schools in these states? (2) What are the main aspects impacting teacher recruitment and retention in Germany? (3) What similarities and differences can be seen?

Taking inspiration from Ball et al. [22], Germany can be regarded as one country, yet different situations and different approaches to deal with the problem of teacher shortage can be described, and policies should be enacted, interpreted, and re-interpreted. The use of enactment theory has been fruitful in highlighting stakeholders' perceptions regarding the phenomenon of teacher shortage. A closer look on the four federal states of Germany and their teaching situation at schools reveals, on the one hand, that local contexts are different, and it is important to have this in mind when dealing with a surplus of educated teachers, as well as with a shortage of teachers. On the other hand, German states cannot act as if they are isolated from the rest of the country. This can be seen in Berlin and Brandenburg in an impressive way. Berlin educates many teachers and, nevertheless, has a great shortage of fully trained teachers, which is partially explainable by the more attractive teaching positions in Brandenburg [50].

The main questions remaining are how successful ministries and politicians as well as the universities are in recruiting the correct number of students for teacher education, and how successful they are in attracting people who are passionate and talented for the teaching profession.

Recruiting teachers usually means employing fully trained teachers; only the best teachers were able to choose their favourite position, but this changed in recent years. Schools and school ministries have difficulties filling their vacant positions. This means

that alternative ways to enter a teaching position should be searched. Some measures, such as employing side entrants, have become established over the years, and are suspended or offered depending on teacher supply. Other measures address the universities; they are asked to offer more study capacities in their Schools of Education. However, at some locations, it is not enough to increase the number of study places—applicants must also be attracted to them—which leads back to the question above. Education policy is asked to find solutions. Making all school types attractive by paying better in primary and lower secondary education is one option, changing the legal status of teachers from an employee to a civil servant is another (cp. Berlin). In addition, the teaching conditions need to be taken into account. School buildings, technical infrastructure, the number of pedagogical assistants and the number of students per class are just some of the areas that urgently need to be looked at.

Compared to other countries, retaining teachers does not seem to be a big issue in Germany. Of course, there are teachers leaving schools, i.e., to work at universities in order to focus on their scientific interests and/or to engage in teacher training, to work in administrative jobs at a school department or in the ministry of education or to do something totally different. However, in general, becoming a teacher means to become a teacher for a lifetime. Universities and ministries focus on recruiting teacher students [51].

Teacher mobility is an issue of utmost importance in Germany. Different teacher education programs do not facilitate movement between the states. Bayern especially does not open easily to graduates from other states. However, this only affects the larger regional movements of teachers. Teacher mobility is also needed when it comes to vacant teaching positions in rural areas. As teachers are rare, they enter the position that they mostly want to teach. This increases the shortage in schools in less attractive regions.

In conclusion, in regard to similarities and differences, the most evident similarity is that all states that have been presented have to deal with a shortage of teachers. Providing education is one of the most important tasks of modern societies, and to fulfilling this task, teachers are needed. How to train the right number of teachers at universities is not easy to say. Fluctuations of student numbers, waves of retirements, and an increased need of new employment may somehow be foreseen but the resources of the universities and of teacher seminars are limited. Additionally, even if more study places at universities are offered, this does not mean that they will be chosen. The differences can be seen in detail. The number of side entrants is remarkably high in Berlin and Brandenburg and remarkably low in Bayern. This may be explainable by the better planning strategies of Bayern or its better recruitment of teachers in the training study places. A call to researchers could be to continue to discuss the results and how these results can be interpreted from the perspective of previous studies and of the working hypotheses. The findings and their implications should be discussed in the broadest context possible.

## 7. Conclusions and Future Research

The exploration of statistical data regarding university graduations and teacher recruitment and retainment provides insights into the phenomenon of teacher shortage on a national and regional level. However, it does not provide a picture of teacher shortage in local schools. Future research, following this initial overview, should strive to deepen the understanding of the phenomenon of teacher shortage in the local school context. For example, school principals, local school administrators, and district-level school administrators could be interviewed and asked about their insights in the challenges and possibilities in regard to teacher shortages. How teachers are recruited and retained will be of interest to study on the local, regional, and national levels in Germany.

**Author Contributions:** Conceptualization, S.S. and M.H.L.; Methodology, S.S. and M.H.L.; Data curation, S.S.; Writing—original draft, S.S.; Writing—review & editing, M.H.L. All authors have read and agreed to the published version of the manuscript.

**Funding:** This research was funded by the Swedish Research Council (2020-04088, WAT's UP What About Teacher Shortage).

**Institutional Review Board Statement:** Not applicable.

**Informed Consent Statement:** Not applicable.

**Data Availability Statement:** All data available online.

**Conflicts of Interest:** The authors declare no conflict of interest.

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
