# Peer review of "Dealing with Teacher Shortage in Germany—A Closer View of Four Federal States"

_education, doi:10.3390/educsci13030227_

Round 1

Reviewer 1 Report

Lines 10-14 = the statement is not clear; can be rephrased for better comprehension; too long for a sentence

Lines 42-43 = what is being compared for similarities/differences among the 4 states in Germany?; clarify

Line 46 = the study has been done, so the tense should be in the “past”

Lines 104-106 = 4 cases are the 4 states? Consider consistent use of “terms”; What similarities/differences are being explored to be specific?

Line 107 = the success of the study depends on the transparency of the methodology, hence, it could be more comprehensive and clear to the readers if the “materials and methods” could follow the sequence of; research design, participants, instruments, data gathering procedure, and data analysis…

Line 111 = what is being referred to “main”?

Line 124 = “particular/istic” is very redundant

Line 131 = results should be aligned with the research questions in terms of presentation of the data, interpretation, and discussion…

Lines 132, 135 = use past “tense”; avoid redundancy

Line 127 = what instrument/s was/were used to obtain data on the different stakeholders’ perspectives (is it not perceptions rather than perspectives?)

Author Response

Please see the attatchment.

Many thanks!

Reviewer 2 Report

The purpose of the manuscript is to analyze differences in dealing with teacher shortages in four representative German countries.  The problem is relevant not only from the point of view of national education policy, but also on an international/European scale, as is made clear in the introduction. The solutions analyzed may therefore be useful in other European countries.  The manuscript is presented in a well-structured manner, the language is clear. The review is clear and comprehensive. It points out the important role of education, especially the key role of the teacher, and the current challenges facing German education (refugee crisis), which are also present in other European countries. The problem presentation, research and discussion form a logical sequence, leading to important recommendations for education policy. A proper theoretical basis for the research - enactment theory - and an appropriate sampling method are used. The analysis of the teacher employment and the effectiveness of the adopted solutions was presented in the socio-geographical context of the state in question, the indicators were selected accordingly. Data are appropriately and consistently interpreted throughout the article. The areas of further research were indicated as well. The bibliography list is up-to-date and comprehensive. I suggest referring in the discusssion to studies by other authors, if any are available.

Reviewer 3 Report

Overview of paper:

The article set out to explore and analyse the teacher supply situation in four German states (Bayern, Berlin, Brandenburg, Nordrhein-Westfalen). More specifically, the author(s) sort to answer three research questions; what is the teacher supply situation in these German states? What factors impact on teacher recruitment and retention in Germany? And what similarities and differences can be seen across the four states? They did this using document analysis (thematic analysis).

The results contribute to the field by drawing attention to differences in the prevalence of shortages within the four identified states. The author(s) also identify distinctions in policy and context that may have contributed to the situation.

Assessment of the paper’s strengths and weaknesses:

As the author(s) makes clear (via discussion and up-to-date academic citations), teacher supply is an area of international concern. Many countries are experiencing teacher supply issues, but the specifics of the problem vary from context to context. A barrage of interventions have been implemented, but none have been shown to be effective in reducing teacher shortages. A wide variety of stakeholders would therefore be interested in reading research that focuses on this area.  

The authors(s) contribute to this evidence-base by assessing the teacher supply situation in Germany. To be more precise, they collate information from various statistical reports and perform a thematic analysis of recent policy documentation. Document analyses are well-suited to describing contextual and historical influences. The chosen research methods are therefore adept at providing the type of information that is required to address their research questions. Readers will be interested to hear about the prevalence of shortages in particular areas and phases of education. Though the article’s clear strength lies in its rich and nuanced description of the factors that may have impacted upon the teachers supply situation within each state and the policy solutions that have been implemented thus far. The paper is also logically structured and provides most of the information that would be required to replicate the review (though a few extra details on the number and type of documentation included within the analysis, would enhance the credibility of the report).

The paper could be improved by including a more extensive review of past research.  This section should clarify what we know and do not know about teacher supply in Germany. Since most analyses of teacher supply tend to focus upon national-level shortages and/or national-level policy interventions, this would help to highlight the unique contribution that the study makes to the field. It would also overtly justify their choice or research method.

Secondly, the author should discuss the strength of their research evidence and acknowledge its limitations. Whilst this paper draws some intriguing connections between particular types of supply issue and state policies and/or contexts (for example, between the employment status of teachers in Berlin, the lack of applications for teaching positions in Berlin, and the quality of education in Berlin), causal associations are notoriously difficult to prove. The author should therefore clarity the evidence that supports their major observations and the level of assurance that we can have in each conclusion.  

Finally, several sections of the paper are unclear. In the opening sections, for example, the author is not explicit about phases of education that the analysis will cover, or the measure of teacher shortages that will be adopted (the number of vacant teaching positions). The reader has to work this out for themselves. Moreover, there are passages of highly repetitive text (see lines 92-102, for example), and instances when the author(s) report information that does not appear to directly relevant to their arguments. For instance, the fact that Berlin sees itself as a “lovable and liveable place” (ln 176), the percentage of Berlin that is marked for woodland, agricultural areas, water bodies or parks (ln 182) or the fact it has 1800 playgrounds (ln 184). This all serves to distract the reader and detracts from their initial impression of the paper. The paper would also benefit from a thorough proof reading to correct for spelling errors, grammatical mistakes, the miss-labelling of section headings (these do not appear in numerical order) and any inconsistencies between the figures cited within Table 1 and the figures cited within-in the text (in the cases where the data comes from different sources, this should be made clear).
